

# Microbial signature profiles of *Penaeus vannamei* larvae in low-survival hatchery tanks affected by vibriosis

Guillermo Reyes[1], Betsy Andrade[1], Irma Betancourt[1], Fanny Panchana[1], Ramiro Solórzano[1], Cristhian Preciado[1], Lita Sorroza[2], Luis E. Trujillo[3] and Bonny Bayot[1,4]

[1] Centro Nacional de Acuacultura e Investigaciones Marinas, CENAIM -ESPOL, Escuela Superior Politécnica del Litoral, ESPOL, Guayaquil, Ecuador
[2] Facultad de Ciencias Agropecuarias, Universidad Técnica de Machala, 5.5 Av Panamericana, Machala, Ecuador
[3] Industrial Biotechnology Research Group, CENCINAT, Universidad de las Fuerzas Armadas, ESPE, Sangolquí, Ecuador
[4] Facultad de Ingeniería Marítima y Ciencias del Mar (FIMCM), Escuela Superior Politécnica del Litoral, ESPOL, Guayaquil, Ecuador

## ABSTRACT

Vibriosis is caused by some pathogenic *Vibrio* and produces significant mortality in Pacific white shrimp *Penaeus (Litopenaeus) vannamei* larvae in commercial hatcheries. Acute hepatopancreatic necrosis disease (AHPND) is an emerging vibriosis affecting shrimp-producing countries worldwide. Zoea 2 syndrome is another type of vibriosis that affects the early stages of *P. vannamei* larvae. Although the pathogenesis of AHPND and zoea 2 syndrome is well known, there is scarce information about microbial composition and biomarkers of *P.vannamei* larvae affected by AHPND, and there is no study of the microbiome of larvae affected by zoea 2 syndrome. In this work, we characterized the microbiome of *P. vannamei* larvae collected from 12 commercial hatchery tanks by high-throughput sequencing. Seven tanks were affected by AHPND, and five tanks were affected by zoea 2 syndrome. Subsequently, all samples were selected for sequencing of the V3–V4 region of the16S rRNA gene. Similarity analysis using the beta diversity index revealed significant differences in the larval bacterial communities between disease conditions, particularly when *Vibrio* was analyzed. Linear discriminant analysis with effect size determined specific microbial signatures for AHPND and zoea 2 syndrome. *Sneathiella*, *Cyclobacterium*, *Haliea*, *Lewinella*, among other genera, were abundant in AHPND-affected larvae. Meanwhile, *Vibrio*, *Spongiimonas*, *Meridianimaribacter*, *Tenacibaculum*, among other genera, were significantly abundant in larvae affected by zoea 2 syndrome. The bacterial network at the phylum level for larvae collected from tanks affected by AHPND showed greater complexity and connectivity than in samples collected from tanks affected by zoea 2 syndrome. The bacterial connections inter *Vibrio* genera were higher in larvae from tanks affected by zoea 2 syndrome, also presenting other connections between the genera *Vibrio* and *Catenococcus*. The identification of specific biomarkers found in this study could be useful for understanding the microbial dynamics during different types of vibriosis.

Corresponding authors
Guillermo Reyes,
guianrey@espol.edu.ec
Bonny Bayot, bbayot@espol.edu.ec

# INTRODUCTION

The genus *Vibrio* is a component of crabs, mollusks, and shrimp microbiomes (*Gao et al., 2019*). Some *Vibrio* species are pathogenic and cause various diseases in aquaculture, commonly referred to as "vibriosis" (*Egidius, 1987*; *de Souza Valente & Wan, 2021*). Vibriosis has a negative economic impact, especially in the culture of Pacific white shrimp *Penaeus (Litopenaeus) vannamei* (*Abdel Latif et al., 2022*). The most common vibriosis in culture shrimp are luminescent vibriosis, acute hepatopancreatic necrosis disease (AHPND), shell disease syndrome, tail necrosis, septic necrosis, zoea 2 syndrome aka bolitas syndrome, red body disease, adult vibrio-caused bacteremia, summer syndrome, seagull syndrome, among others (*Vandenberghe et al., 1999*; *de Souza Valente & Wan, 2021*; *Abdel Latif et al., 2022*).

Zoea 2 syndrome is a vibriosis caused by *Vibrio alginolyticus* and *V. harveyi,* that affects the early stages of *P. vannamei* larvae (*Robertson et al., 1998*; *Vandenberghe et al., 1999*; *Kumar et al., 2017*). AHPND is one of the most emerging vibriosis and continues to cause significant losses to shrimp producers worldwide (*de Souza Valente & Wan, 2021*). The main causative agent of AHPND is *V. parahaemolyticus*, which contains a pVA1 plasmid carrying two toxin genes (*PirAB*). *PirAB* plasmids can be transferred between *Vibrio* species, diversifying the pathogenic bacteria plethora (*Xiao et al., 2017*; *Liu et al., 2018*; *Dong et al., 2019*). AHPND bacteria replicate in the shrimp stomach and consequently invade the hepatopancreas causing its degradation, which finally kills the shrimp (*Han et al., 2015*). The disease can affect shrimp at any stage causing mortalities up to 100% (*Leobert et al., 2015*).

The virulence factors of AHPND-causing bacteria cause perturbations in the shrimp microbiome promoting the development of the disease (*Zhang et al., 2021*). Relevant microbiome studies have identified biomarkers associated with the pathogenic AHPND-causing *V. parahaemolyticus* (*Restrepo et al., 2021*), non-AHPND *V. parahaemolyticus* (*Zhang et al., 2021*), and *V. alginolyticus* (*Liao et al., 2022*) in juvenile *P. vannamei* shrimp through experimental studies using challenge tests. For example, the genera, *Pseudoalteromonas*, *Bacterioborax*, *Ruegeria*, and other members of the Vibrionales order were the most abundant in juvenile shrimp challenged with a *Vibrio* strain causing AHPND (*Restrepo et al., 2021*). *Oceanospirillum*, *Lewinella*, and *Saprospira* were the most abundant genera in juvenile shrimp challenged with a non-AHPND *V. parahaemolyticus* strain (*Zhang et al., 2021*). *Pseudomonas*, *Candidatus_Bacilloplasma*, and *Shewanella* were the most abundant genera in juvenile shrimp challenged with a strain of *V. alginolyticus* (*Liao et al., 2022*). On the other hand, the microbiomes of healthy and juvenile-farmed *P. vannamei* shrimp naturally affected by AHPND were also compared through observational studies (*Chen et al., 2017*; *Cornejo-Granados et al., 2017*; *Dong et al., 2021*). *Vibrio* was abundant in juvenile shrimp naturally affected by AHPND (*Chen et al., 2017*; *Dong et al., 2021*).

*Candidatus Bacilloplasma* (*Dong et al., 2021*) and *Aeromonas*, *Simiduia*, and *Photobacterium* (*Cornejo-Granados et al., 2017*) were also abundant in shrimp naturally affected by AHPND.

Little research has been conducted on microbial biomarkers of AHPND in *P.vannamei* larvae, and there is no study of the microbiome of larvae affected by zoea 2 syndrome. In one of the first studies on the health of *P. vannamei* and its microbiome, (*Zheng et al., 2017*) conducted an observational study to establish disease biomarkers (*Nautella* and *Kordimonas* genera) using external signs as criteria to classify the disease group. In a previous study, we found microbial biomarkers associated with high and low survival of *P. vannamei* larvae naturally affected by AHPND in commercial hatchery tanks, for the future development of probiotics to mitigate the disease (*Reyes et al., 2022*). *Catenococcus*, *Gilvibacter*, *Sneathiella*, and *Marinibacterium*, among other genera, decreased the survival of AHPND-affected postlarvae, whereas *Bacillus*, *Mameliella*, *Bdellovibrio*, and *Yangia*, among other genera, increased the survival of AHPND-affected postlarvae (*Reyes et al., 2022*). However, there are no other observational or experimental studies of biomarkers in *P. vannamei* larvae affected by AHPND or zoea 2 syndrome.

Considering the research gap in understanding the microbial dynamics associated with different vibriosis in *P. vannamei* larvae, more research on microbial biomarkers is needed. In this context, we conducted an observational study of the microbiome of *P. vannamei* larvae in a commercial hatchery during a critical period when all tanks were affected by two types of vibriosis: AHPND and zoea 2 syndrome, resulting in most tanks having low survival at harvest, and no disease-free tanks were detected. The aim of this study was therefore to compare the microbial profiles of larvae naturally affected by these two types of vibriosis: AHPND and zoea 2 syndrome. This is the first report on the microbiome of larvae affected by zoea 2 syndrome and a comparison of the microbial profiles of larvae affected by two vibriosis in commercial hatchery tanks. The results of this study are useful for understanding the microbial dynamics during different vibriosis diseases.

## MATERIALS AND METHODS

### Ethics statement

No ethics statement or permissions were required for this study. However, in order to ensure the ethical treatment of animals, a two-step anesthesia-euthanasia protocol of animal submersion in 60% and 95% alcohol was used to sacrifice them.

### Sources of samples and microbiome data

An observational study was conducted in a commercial shrimp hatchery (64 tanks, 35 tons, South America) to investigate the microbiome of *P. vannamei* larvae naturally affected with vibriosis (Fig. S1). Two investigations were generated from the samples collected in the observational study (Fig. S1). In the first investigation, already published by the authors of this article (*Reyes et al., 2022*), the microbiome of larvae collected from high (22 samples from seven tanks) and low (25 samples from other seven tanks randomly selected from 51 tanks) survival tanks affected by AHPND was compared for future development of probiotics to mitigate AHPND (Fig. S1). In the second research (this article), we re-used the microbiome data obtained from the low survival AHPND-affected tanks of our first study
(accession number PRJNA800805). This data was then compared with the microbiome data from zoea 2 syndrome-affected larvae (20 samples from five different tanks, microbiome data presented for the first time in this article) with the new aim of comparing the microbial profiles of larvae affected by two types of vibriosis: AHPND and zoea 2 syndrome (Fig. S1).

## Sample collection

Samples of larvae of the following stages were collected from each one of the 12 tanks: Mysis 3 (M3), Postlarvae 4 (PL4), Postlarvae 7 (PL7), and Postlarvae 10 (PL10), except in two of the seven tanks affected by AHPND, where only samples for M3, PL4, and PL7 were collected in each of the two tanks because the populations died at the PL9 stage (Fig. S1). In addition, a sample (M3) from a third tank affected by AHPND failed DNA quality control (Fig. S1). In conclusion, a total of 45 samples of *P. vannamei* larvae from the 12 commercial hatchery tanks (25 and 20 samples from tanks affected by AHPND and zoea 2 syndrome) were used for the microbiome analysis of this article (Fig. S1). Survival [mean ± standard error of the mean (SEM)] at harvest from the 12 tanks was 24.4 ± 5.2%. In addition, observations for clinical signs of disease were made at each sampling (5 to 7 min) at the larval stages PL7 and PL10. Ranges of water quality parameter were pH = 7.9–8.1, salinity = 35 g/L, dissolved oxygen concentration = 5.0 mg/L, and temperature = 32.5–33 °C. Larvae were supplemented with four commercial diets (Skretting PL, Frippak Fresh, Zeigler Z Plus, and Advance Brine Shrimp).

## Sample processing and high-throughput sequencing

The samples were processed for three kinds of analysis: (1) Polymerase chain reaction (PCR) analysis to detect bacterial infections caused by AHPND (detection of *PirAB* toxin genes) and discard infections with common shrimp pathogens (white spot syndrome virus–WSSV, infectious hypodermal and hematopoietic necrosis virus–IHHNV, *Enterocytozoon hepatopenaei*–EHP, and shrimp hemocyte iridescent virus–SHIV), (2) histological analysis to confirm the presence of lesions compatible with bacterial infections, and discard lesions caused by other common shrimp pathogens, and (3) microbiome analysis through *16S rRNA* amplicon high-throughput sequencing (HTS).

Approximately, 1 g of each sample (M3, PL4, PL7, and PL10) was washed with a 2% NaCl sterile solution to remove external impurities, macerated, and divided into two aliquots. The first aliquots of macerates (500 μl) for each sample were processed for PCR analysis. The genomic DNA (gDNA) extraction was carried out using the phenol-chloroform protocol. The *PirAB* toxin genes were amplified with the AP4 F1/R1 - AP4 F2/R2 primers (*Dangtip et al., 2015*). WSSV, IHHNV, EHP, and SHIV infections were diagnosed with the IK1/2-IK3/4, IHHNV309F/R, SWP 1F/1R-SWP 2F/2R, and SHIV-F1/R1-SHIV-F2/R2 primers, respectively (*Lo et al., 1996*; *Tang, Navarro & Lightner, 2007*; *Jaroenlak et al., 2016*; *Qiu et al., 2017*).

Twenty larvae of each sample collected at the PL7 and PL10 stages were preserved in Davidson's fixative solution for histological analysis. Shrimp tissues were processed according to a previously published protocol where the sections were cut at 5 um and stained with Mayer-Bennet haematoxylin and eosin (*Bell, 1988*).

The 45 samples were allocated into two groups according to the disease condition: affected by AHPND: samples collected from tanks where AHPND was detected by PCR (at any larval stage) and histological analysis (only for PL7 and PL10 larval stage), and affected by zoea 2 syndrome: samples from tanks where AHPND was negative by PCR (at any larval stage) and also by histological analysis (only for PL7 and PL10 larval stage), but histological lesions were compatible with zoea 2 syndrome.

The second aliquots of macerates (500 µl) obtained from each sample were distributed into 2-mL cryovial tubes for a snapshot freezing with liquid nitrogen ($-196$ °C) and immediately stored at $-80$ °C for further microbiome analysis and its functional prediction. These macerates remained at $-80$ °C until all samples were collected at the end of the production cycle of all 12 tanks. Then, all macerates were processed for gDNA extraction using the ZymoBiomics DNA Microprep Isolation Kit (Zymo Research, Irvine, CA, USA), following the manufacturer's instructions. A 20 µl aliquot of gDNA was diluted in DNase-free ultrapure water, and DNA quality and concentration (A260/280 $= 1.8-2.0$) were examined through a NanoDrop One Microvolume UV-Vis Scanning Spectral Spectrophotometer (Thermo Fisher Scientific, Waltham, MA, USA). The gDNA of all 45 samples was then submitted to Novogene Incorporation (Beijing, China) for amplification of the V3–V4 region of the *16S rRNA* gene using the 341F/806R primers (*Takahashi et al., 2014*), amplicon library construction, and paired-end sequencing with the Illumina NovaSeq 6000 P250 platform. The DNA sequences of the 20 samples from the five tanks affected by zoea 2 syndrome were deposited at the National Center for Biotechnology Information (NCBI) Sequence Read Archive (SRA) under accession number PRJNA870476. The sequences of the 25 samples from the seven tanks affected by AHPND are available under accession number PRJNA800805 (*Reyes et al., 2022*).

## Sequence analysis

Sequencing reads were processed with the Divisive Amplicon Denoising Algorithm (DADA2) version 1.16 (*Callahan et al., 2016*). Sequences were filtered with the filterAndTrim function to obtain reads higher than Q30. Sequences with more than two expected errors were removed [maxEE $=$ c (2,2)], and the last 10 nt of the forward reads and the last 20 nt of the reverse reads were trimmed [truncLen $=$ c (240,230)]. The filtered reads were dereplicated, denoised, and merged with the forward and reverse amplicon sequences. Chimeric sequences were then removed using the removeBimeraDenovo function. The table of amplicon sequence variants (ASVs) was generated, and the taxonomy was assigned using the public database SILVA version 138.1 (*McLaren Michael & Callahan Benjamin, 2021*).

## Data analysis

Larvae survival at harvest from tanks affected by AHPND and zoea 2 syndrome were compared by the non-parametric Wilcoxon-Mann–Whitney test using the stats package in R.

The variability of the alpha diversity indices and the relative abundance of the microbiome were studied for the samples collected at the four larval stages of tanks

 

affected by AHPND and zoea 2 syndrome. Shannon and ACE indices were used to estimate the alpha diversity (*Shannon & Weaver, 1949*; *Hughes et al., 2001*). Differences in Shannon and ACE indices, and relative bacterial abundance at family and genus levels between the disease conditions were evaluated through the non-parametric Wilcoxon-Mann–Whitney test.

The microbiome structure, and the *Vibrio* communities among larval samples (beta diversity) collected from tanks affected by the two diseases condition was estimated by the multivariate analysis of the Bray–Curtis dissimilarity index and visualized by principal coordinate analysis (PCoA). The null hypothesis of equality of microbiomes and *Vibrio* communities between samples collected from tanks affected by AHPND and zoea 2 syndrome was evaluated by a similarity analysis (ANOSIM). All previous statistical analyses were considered significant with $P < 0.05$.

Co-occurrence network analyses at the phylum and genus levels were performed to evaluate the interactions among bacterial taxa at each disease condition. The co-occurrence network was built through a Spearman correlation matrix of the ASV abundances using the trans_network$new function. Co-occurrence was considered robust when positive or negative correlation coefficients were greater than 0.6 and $P < 0.01$ (*Barberán et al., 2012*). The cal_sum_links function was used to calculate for each ASV (edge) the number of other ASVs with significant correlations ($P < 0.01$) to the correspondent edge and plotted with the plot_sum_links function.

Differential abundance analysis between the disease conditions was evaluated through a linear discriminant analysis (LDA) of the ASVs with effect size (LEfSe) (*Segata et al., 2011*). ASVs with an LDA cutoff of 2.0 and significant differences ($P < 0.05$) between both disease conditions were considered potential microbial signatures. The robustness of the microbial signatures obtained with the LEfSe analysis was evaluated through a Bayesian *t*-test analysis using the BayesFactor package (*Morey & Rouder, 2011*). The Bayes factor $BF_{10}$ was calculated as the ratio of the probability of $H_1$ (different abundance of ASVs between both disease conditions) over the probability of $H_0$ (equal abundance of ASVs between both disease conditions) based on the data. $BF_{10}$ with values: 1–3, 3–10, and >10 showed weak, moderate, and strong evidence in favor of $H_1$ (*Morey & Rouder, 2011*; *van Doorn et al., 2021*).

The ability of the bacterial functionality was predicted using the *16S rRNA* amplicon data against the pathway reference profiles (Ref99NR) of the Kyoto Encyclopedia of Genes and Genomes (KEGG) database (*Kanehisa et al., 2012*). Tax4fun2 was used to predict the functionality of the ASVs at hierarchical levels 1, 2, and 3 (*Wemheuer et al., 2020*). ASVs with similarities higher than 97% to the reference database were considered Orthologous KEGG groups (KOs). A differential abundance analysis between both disease conditions was performed to identify significant ($P < 0.05$) functional biomarkers through LEfSe analysis of the KOs, with an LDA cutoff of 2.0. All downstream analyses, except the Bayesian analysis, were performed with the microeco package (*Liu et al., 2021*) in the R software version 4.2.1 (*R Core Team, 2022*).

## RESULTS

After *PirAB* toxin genes amplification by PCR, a total of 25 samples collected from seven tanks were AHPND-positives and the remaining 20 samples collected from the other five tanks were AHPND-negatives. All 45 samples were negatives to WSSV, IHHNV, EHP, and SHIV.

Histological analysis confirmed the presence of AHPND lesions by severe necrosis of hepatopancreatic tubules at the terminal phase in larval samples collected at PL7 and PL10 stages from the seven tanks that were AHPND positive by PCR analysis (Figs. 1C and 1D). On the other hand, the larvae samples collected at the PL7 and PL10 stages from the other five tanks showed no lesions provoked by AHPND (Figs. 1E and 1F). Rather, these larvae presented lesions in the hepatopancreas tubules: vacuolization and detachment of their cells into the stomach lumen, and hemocytic infiltration in the connective tissue (Fig. 1E). Also, it was observed detachment of hindgut epithelial cells into its lumen and vacuolization in the intestine (Fig. 1F). These lesions were compatible with zoea 2 syndrome. In any of the 45 samples were observed lesions compatible with septic vibriosis or caused for any virus. Larvae collected in all tanks at the PL7 and PL10 stages presented abnormal swimming behavior (erratic swimming and lethargy), delayed growth, and pale atrophied hepatopancreas. In addition, all samples collected from the five tanks affected by zoea 2 syndrome also showed empty digestive tracts (PL7 and PL10) and body color changes to off-white.

Non-significant difference in survival ($P = 0.150$) was observed between the seven AHPND-positive tanks (mean $\pm$ SEM $= 17.9 \pm 4.6\%$) and the five tanks affected by vibriosis (mean $\pm$ SEM $= 33.7 \pm 9.9\%$) at harvest (Table S1).

### Sequence analysis

The sequence analysis of the 45 samples generated approximately six million clean reads, with an average of 146,311 clean reads per sample (Table S1). The error rate across the 45 samples averaged 0.03% for all the clean reads (Table S1). On average, 88.06% of these reads exhibited a Phred score Q >30 and a GC content of 53.72% (Table S1). Amplicons were clustered into 8,617 ASVs, with an average of 392 ASVs per sample (Table S1). Good's coverage was higher than 99.99%, indicating an optimal sequencing depth (Table S1).

### Microbiome composition and statistical analysis

No significant differences in the Shannon and ACE indices of the larvae microbiome were found between disease conditions ($P = 0.876$ and 0.100, Fig. S2). The distribution of the relative abundance of the microbiome showed the prevalence of 17 bacterial phyla, representing 29 classes, 75 orders, 105 families, and 182 genera in all larval samples.

At the phylum level, Pseudomonadota exhibited the highest and most similar abundance in tanks affected by AHPND (72%) and zoea 2 syndrome (77%) (Table S2 and Fig. S3). Bacteroidota (16%) and Bacillota (3%) were the following phyla with the highest global average abundances (Table S2 and Fig. S3).

At the family level, Rhodobacteraceae, Vibrionaceae, and Flavobacteraceae exhibited the highest abundance at both AHPND and zoea 2 syndrome-affected tanks (Fig. 2 and

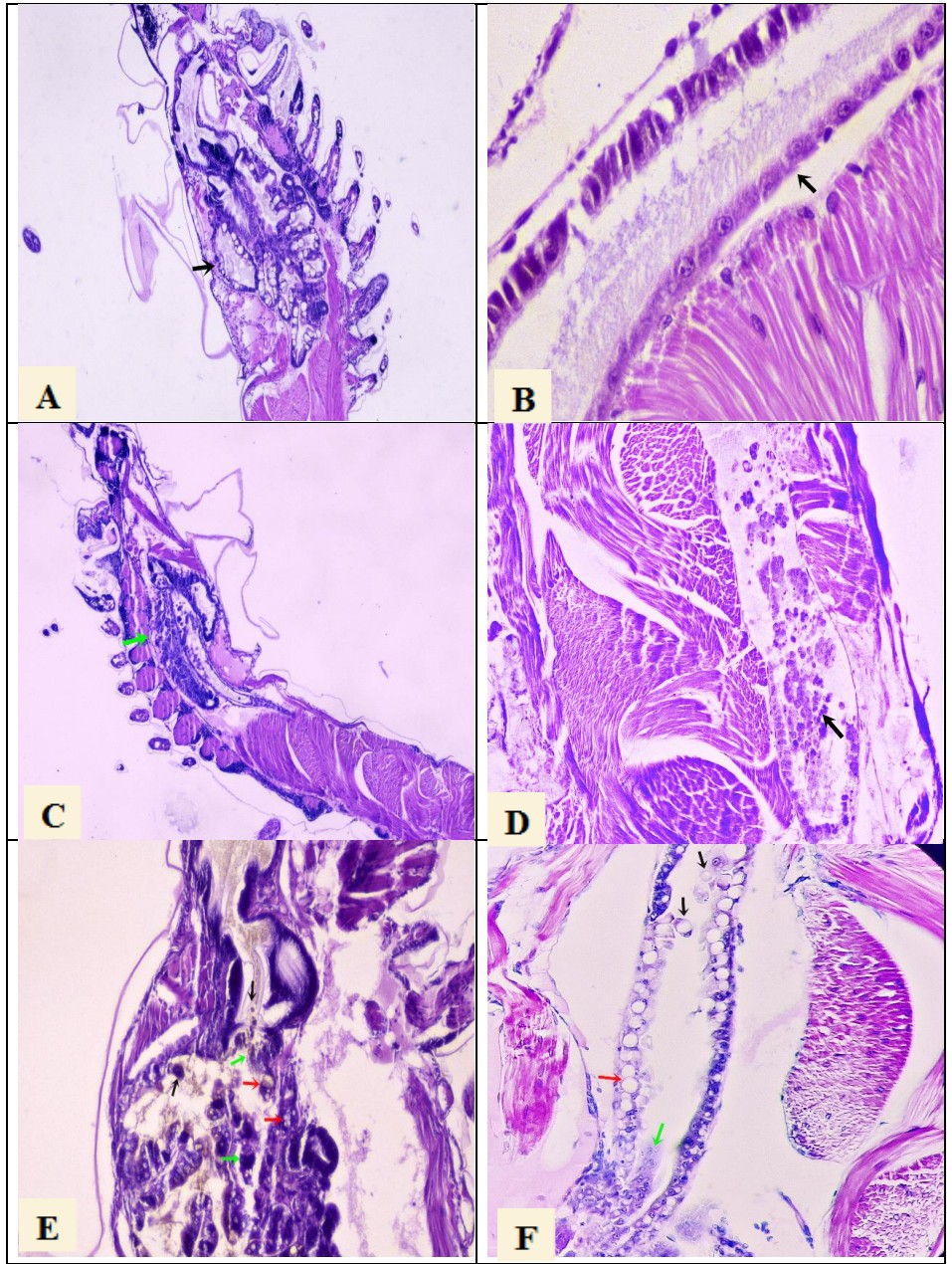

**Figure 1** **Histological section of *P. vannamei* shrimp postlarvae stained with Mayer-Bennet haematoxylin and eosin.** (A) The structure of the tubules of the hepatopancreas is observed under normal conditions, black arrow, 4x. (B) The midgut epithelium is observed in normal conditions, black arrow, 10x. (C) Necrosis of the hepatopancreatic tubules show a terminal phase of AHPND with the destruction of epithelial cells, green arrow. 4x. (D) Necrosis and detachment of epithelial cells of the hindgut into its lumen, black arrow. 10x. (E) Hepatopancreatic tubules show cell detachment, black arrows, necrosis of the tubules, green arrows, and vacuolization in the tubules caused by bacterial infection, red arrow. 10x. (F) Detachment of hindgut epithelial cells into the lumen, black arrows, necrosis in the epithelium, green arrow, and vacuolization due to bacterial infection, red arrow. 10x.

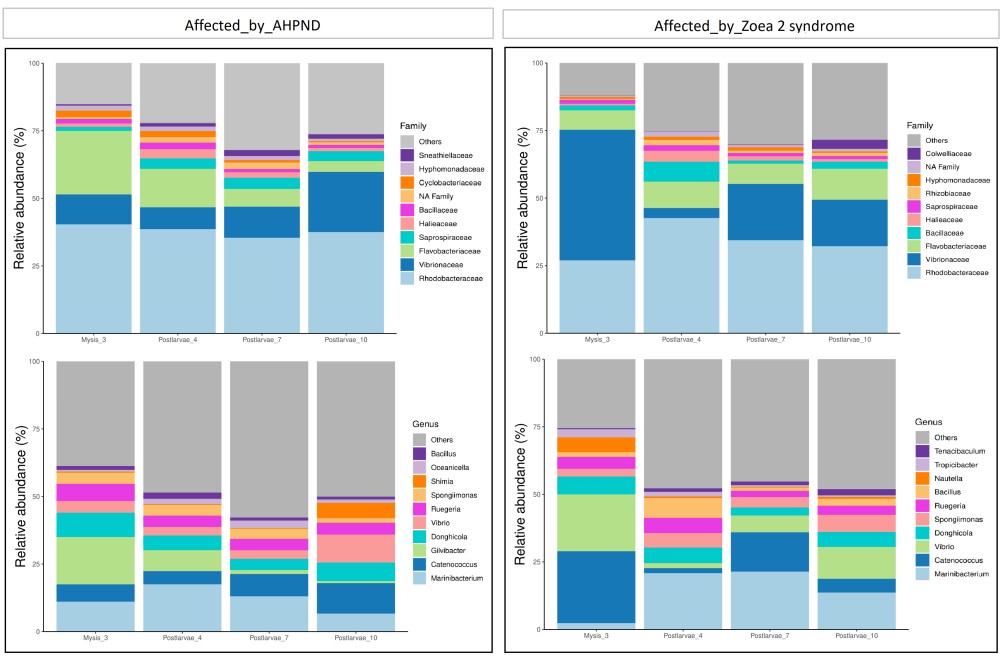

**Figure 2 Relative abundance of the larval microbiome of samples collected from tanks affected by AHPND and zoea 2 syndrome.** (A) Family level. (B) Genus level.

Table S2). Vibrionaceae exhibited almost a two-fold increase of abundance ($P = 0.015$) in tanks affected by zoea 2 syndrome, especially at the M3 stage (49%), in contrast to the same larvae stage in AHPND-affected tanks (11%) (Table S2). Flavobacteraceae was most abundant at the M3 stage from AHPND-affected tanks, being three-fold more abundant than at the same larval stage from tanks affected by zoea 2 syndrome, but overall was not significantly different between the two diseases ($P = 0.120$) (Fig. 2 and Table S2).

At the genus level, *Marinibacterium* exhibited the highest abundance, especially at the PL4 and PL7 (22%) stages of tanks affected by zoea 2 syndrome (Fig. 2B and Table S2). *Catenococcus* and *Vibrio* were the second and third most abundant genera, respectively, in tanks affected by zoea 2 syndrome, with almost a two-fold increase compared to the corresponding AHPND cohort ($P = 0.016$, and $P = 0.013$, respectively) (Fig. 2B and Table S2). *Gilvibacter* was seven-fold more abundant ($P = 0.044$) in AHPND-affected tanks, especially at the M3 stage (18%). *Bacillus* was the least abundant genus in larvae at both disease conditions (Table S2).

A notable difference was observed when bacterial communities (beta diversity) were compared between the two disease conditions. The ANOSIM analysis of the beta diversity showed a remarkable separation of the larval microbiome affected with both disease conditions ($P = 0.005$, Fig. 3A). Moreover, significant differences in the shrimp microbiome were observed between disease conditions at M3 ($P = 0.005$), PL7 ($P = 0.039$), and PL10 ($P = 0.040$) stages, except at PL4 ($P = 0.130$). Closer analysis of the *Vibrio* genus revealed a significant separation ($P = 0.001$) between the two diseases (Fig. 3B).
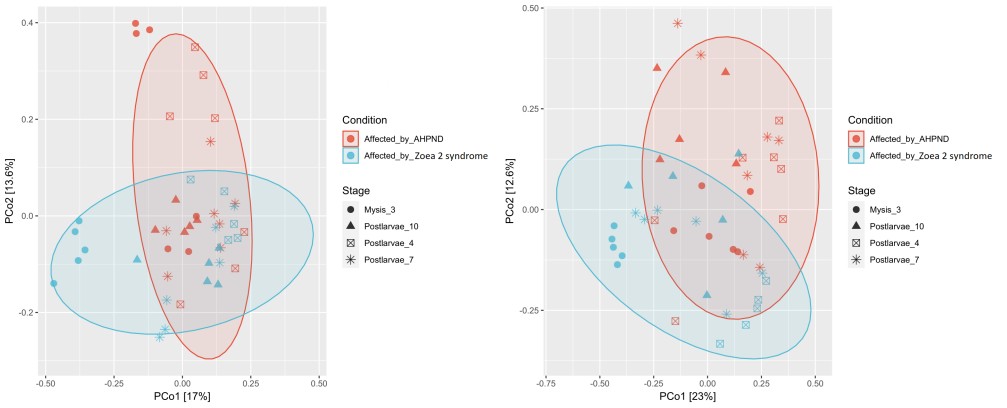

**Figure 3** **Principal coordinate analysis (PCoA) of the Bray-Curtis dissimilarity index of the *P. vannamei* larval microbiome collected from tanks affected by AHPND and zoea 2 syndrome.** Significant differences (ANOSIM, $P < 0.05$) in the (A) microbiome structure and (B) *Vibrio* communities were observed between disease conditions.

The co-occurrence bacterial network at the phylum level for the samples collected from tanks affected by AHPND showed a higher complexity and connectivity than in samples collected from tanks affected by zoea 2 syndrome (Fig. 4, Table S3). This higher complexity of the microbial communities of samples collected from tanks affected by AHPND was reflected by higher values (between 20% and 2.2-fold higher) of the number of edges, network centralization, density, clustering coefficient, heterogeneity, and average degree, and a lower average path length (Table S3). Such complexity was observed despite the bacterial network for both disease conditions showed the same network diameter, and only a 0.3% higher number of nodes for the bacterial network of samples collected from tanks affected by zoea 2 syndrome (Fig. 4, Table S3).

Pseudomonadota was directly associated with 251 edges at the phylum level in samples affected by AHPND (Fig. 4). A total of 112 of these edges were connected to ASVs of the same Pseudomonadota phylum, 105 edges were connected to Bacteroidota ASVs, and the remaining 34 edges were connected to ASVs of the following phylum: Desulfobacterota, Bacillota, Myxococcota, Campylobacterota, Actinomycetota, Bdellovibrionota, Chloroflexi and Cyanobacteria (Fig. 4). In samples affected by zoea 2 syndrome, Pseudomonadota was directly associated with 220 edges, of which 146 were connected to ASVs of the same Pseudomonadota phylum, 41 edges were connected with Bacteroidota ASVs, and the remaining 33 edges were connected with ASVs of the following phylum: Actinomycetota, Desulfobacterota, Bdellovibrionota, Bacillota, Myxococcota, Chloroflexi and Patescibacteria (Fig. 4).

The network analysis at the genus level showed a high number of ASVs without taxonomic identification at both disease conditions (Figs. S4, S5). In the co-occurrence network of the AHPND-affected larvae, *Vibrio* edges were mostly associated with *Catenococcus* (Fig. S4), whereas 23 of the 38 edges in the co-occurrence network of the samples affected by zoea 2 syndrome were associated with ASVs belonging to the
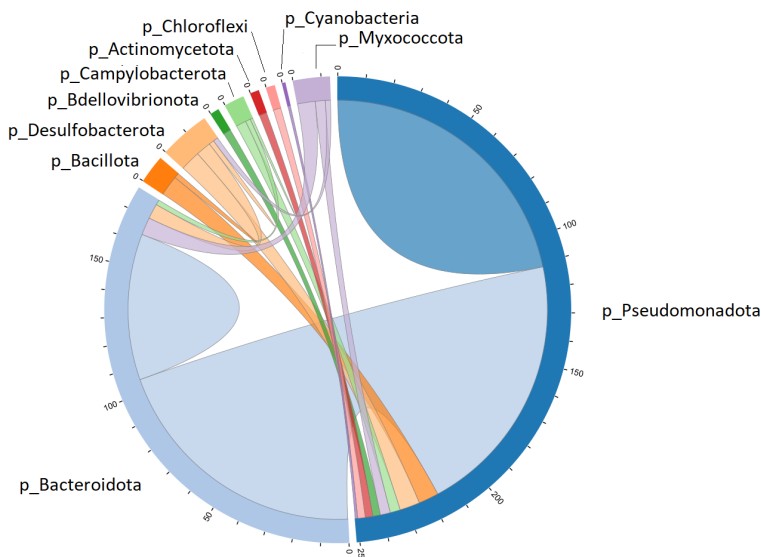

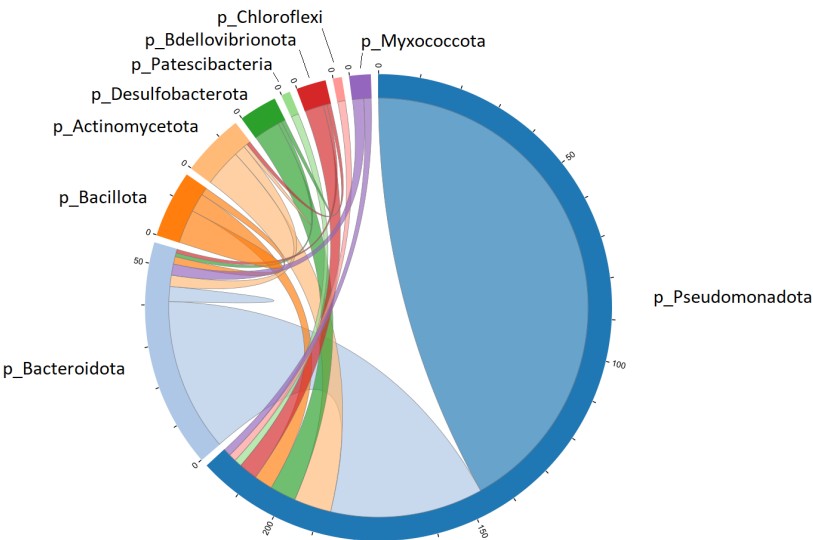

**Figure 4** **Co-occurrence network of bacterial communities present in the larval microbiome of samples collected from tanks affected by AHPND and zoea 2 syndrome.** The numbers along the outside of the pie chart represent the number of ASVs that are related to the phylum. An edge indicates a strong (Spearman correlation > 0.6) and significant ($P < 0.01$) correlation.

same *Vibrio* genus, being this number of edges 2.5-fold higher in larvae affected by zoea 2 syndrome compared to larvae affected by AHPND (Figs. S4, S5).
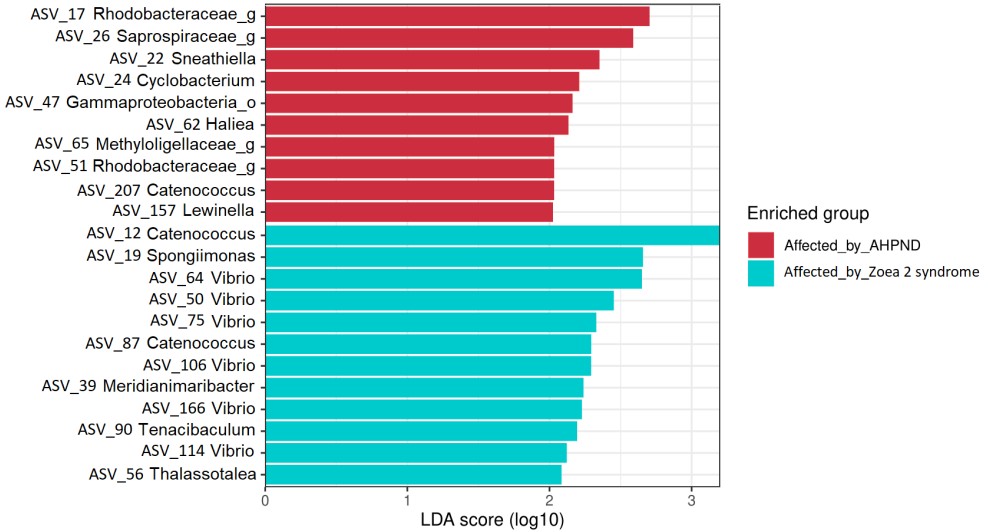

**Figure 5** Results of the linear discriminant analysis (LDA) with effect size (LEfSe) of the larval microbiome of samples collected from tanks affected by AHPND and zoea 2 syndrome. The length of the bar represents the effect size (LDA cutoff = 2) of all bacterial lineages at the genus level.

The LEfSe analysis at the genus level identified 22 significant ASVs, ten of which were significantly more abundant for AHPND-affected larvae (Table S4 and Fig. 5). The genera *Sneathiella*, *Cyclobacterium*, *Haliea*, *Catenococcus*, and *Lewinella* were abundant in AHPND-affected larvae, with *Sneathiella* being the most abundant (Fig. 5). On the other hand, the genera *Catenococcus*, *Spongiimonas*, *Vibrio*, *Meridianimaribacter*, *Tenacibaculum*, and *Thalassotalea* were abundant in the tanks affected by zoea 2 syndrome (Table S4 and Fig. 5). The Bayesian analysis confirmed the evidence (strong and moderate) of the differential abundance of ASVs found through the LEfSe analysis (Table S5). However, two ASVs from tanks affected by zoea 2 syndrome (ASV 56 and ASV 114), corresponding to the *Thalassotalea* and *Vibrio* genera showed weak evidence of differential abundance between the two disease conditions (Tables S4, S5).

A total of 22 bacterial pathways were differentially represented in the comparison analysis between disease conditions using the LEfSe analysis ($P < 0.05$) (Fig. 6). At a higher level (KEGG level 3) the function pathway (metabolism-related) was significantly more abundant in samples affected by AHPND, while cellular processes and environmental information processing were significantly more abundant in samples affected by zoea 2 syndrome (Fig. 6). At lower depth levels (KEGG levels 1 and 2) the following functional pathways were significantly more abundant in AHPND-affected tanks: amino acid and carbon metabolisms, metabolic pathways, biofilm formation, translation, two-component system, signal transduction, global and overview maps, biosynthesis of antibiotics, amino acids, and secondary metabolites (Fig. 6). In contrast, cell motility, phosphotransferase system, quorum sensing, cellular community—Prokaryotes, ABC transporter, and membrane

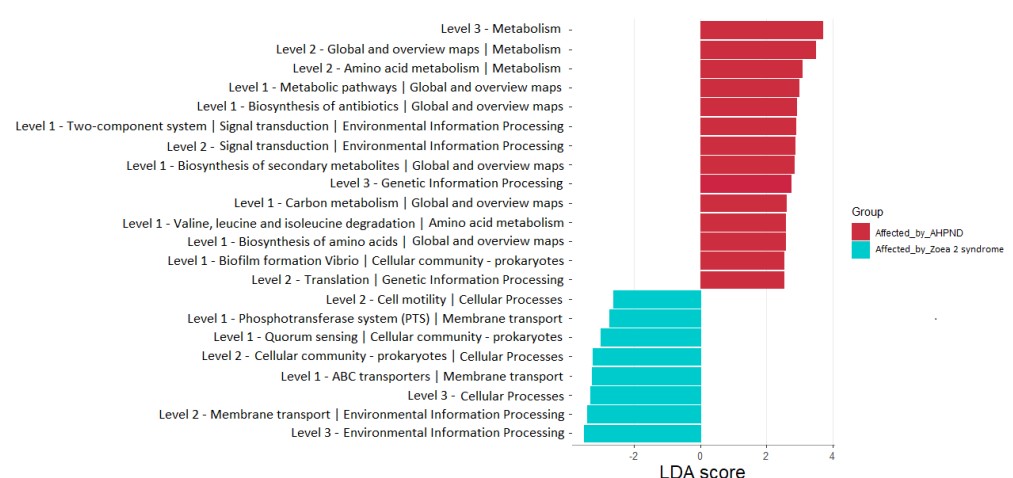

**Figure 6** Functional pathways indicate differences between the microbiome of samples collected from tanks affected by AHPND and other zoea 2 syndrome. Only pathways with an LDA score of more than two after linear discriminant analysis effect size were considered functional biomarkers.

transport were significantly more abundant in samples affected by zoea 2 syndrome (Fig. 6).

# DISCUSSION

AHPND and zoea 2 syndrome are two of the most common vibriosis of *P. vannamei* shrimp. However, information on microbial composition and possible biomarkers of AHPND-infected *P. vannamei* larvae is scarce. Likewise, no study regarding the microbiome of larvae affected by zoea 2 syndrome has been published yet. In the present study, we compared the microbial signature profiles of *P. vannamei* larvae collected from low-survival hatchery tanks affected by the two vibriosis. We focused on comparing the diversity, relative and differential abundance, co-occurrence network, and functional composition of the bacterial communities present at both groups to contribute to the understanding of the microbial dynamics during different types of vibriosis.

Vibrionaceae is a family of great interest, as some species, such as *V. harveyi*, *V. campbellii*, *V. brasiliensis*, *V. alginolyticus*, *V. owensii*, *V. inhibens,* and *V. natriegens*, among others, are pathogenic for cultured shrimp (*Lightner, 1996*; *Karunasagar, Otta & Karunasagar, 1998*; *Hong, Lu & Xu, 2016*; *Hasan et al., 2017*; *Sotomayor et al., 2019*; *Zhang, He & Austin, 2020*; *Kumar et al., 2021*; *Li et al., 2021*). Two of these *Vibrio* species: *V. alginolyticus* and *V. harveyi* cause zoea 2 syndrome (*Robertson et al., 1998*; *Kumar et al., 2017*). As we expected, most of the ASVs abundant in the tanks affected by zoea 2 syndrome, were bacteria taxonomically assigned to the Vibrionaceae family (70% = 7/10 ASVs with strong evidence of differential abundance compared with bacteria from samples affected by AHPND). These ASVs belonged not only to the *Vibrio* genera (5/7), as the remaining two ASVs (2/7) were bacteria from the *Catenococcus* genera. *Catenococcus* is an endophytic bacterium,
sometimes associated with algae or sponges to produce bioactive secondary metabolites that attack or debilitate marine organisms (*Achmad et al., 2016*; *Yoghiapiscessa, Batubara & Wahyudi, 2016*; *Guibert et al., 2020*; *Hagaggi & Abdul-Raouf, 2022*). *Catenococcus* is also pathogenic for *Penaeus indicus* shrimp and can be controlled applicating *Bacillus* probiotic (*Patil et al., 2021*). In this work, the co-occurrence network of bacteria abundance from samples affected by zoea 2 syndrome showed robust positive correlations, mostly within the same Pseudomonadota phylum. Specifically, positive correlations of the ASV abundances within bacteria from the same *Vibrio* genus, and between bacteria from the *Vibrio* and *Catenococcus* genera were found, indicating that the latter could be a common resident of the microbiome of *P. vannamei* affected by zoea 2 syndrome.

The remaining 30% of the ASVs differentially abundant in tanks affected by zoea 2 syndrome (3/10 ASVs with strong evidence of differential abundance compared with bacteria from samples affected by AHPND) were taxonomically assigned to the Bacteroidota phylum (Flavobacteraceae family). The Flavobacteraceae family is part of the intestinal microbiome of *P. vannamei* shrimp (*Zheng et al., 2017*; *Dai et al., 2018*; *Reyes et al., 2022*), and some members are microbial markers of vibriosis in *P. vannamei* shrimp (*Zhang et al., 2021*). Additionally, many members of this family promote the inhibition of pathogenic vibrios by providing bioavailable substrates that benefit the nutritional functions of the *P. vannamei* shrimp (*Dai et al., 2018*). On the other hand, pathogenic *Vibrio* is also capable to benefit from bioavailable substrates produced by the Flavobacteraceae bacteria and thus can take advantage of the presence of beneficial bacteria (*Chen et al., 2017*; *Dai et al., 2018*). *Tenacibaculum* and *Aquimarina* are the genera of the Flavobacteraceae family most widely reported as pathogens and causes of outbreaks in aquatic species, including *P. vannamei* shrimp (*Avendaño Herrera, 2009*; *Zhang et al., 2021*; *Hudson & Egan, 2022*). *Tenacibaculum* is also a biomarker of the cotton shrimp-like disease, mainly characterized by damage in the shrimp hepatopancreas and body color changes to off-white (*Zhou et al., 2019*). Off-white color in the shrimp body has also been observed in *P. vannamei* larvae affected by zoea 2 syndrome (*Vandenberghe et al., 1999*; *Kumar et al., 2017*). Consistently, in our study *Tenacibaculum* was significantly abundant in larvae collected from low-survival tanks affected by zoea 2 syndrome, suggesting that *Tenacibaculum* could be also part of the microbial signature for zoea 2 syndrome.

Damage of the hepatopancreas and hindgut (necrosis, detachment of the epithelial cells and vacuolization), accompanied by mass mortalities at the zoea stages, and external signs of body color changes to off-white of shrimp larvae, occur during the zoea 2 syndrome (*Vandenberghe et al., 1999*; *Kumar et al., 2017*). Although larvae of the group affected by zoea 2 syndrome of our study exhibited the characteristic histopathological lesions of the disease, high mortalities were observed when shrimp reached postlarvae stages 7 and 10. Mass mortalities have not been observed when zoea 2 are experimentally infected with $10^7$ CFU/mL of *V. harveyi*, one of the etiological agents of the zoea 2 syndrome (*Robertson et al., 1998*). Therefore, it is possible that the disease started at early stages, and that the affectation continued until larvae reached postlarvae stages, without the presence of mass mortality during the early shrimp stages. The histological damages observed in the hepatopancreas of the larvae did not seem to be associated with septic vibriosis, as internal and external

melanizations, a typical sign of septic vibriosis, were not observed (*Lightner & Lewis, 1975*; *Ruangpan & Kitao, 1991*; *Haldar et al., 2007*).

Even though the ASVs abundant in samples from tanks affected by AHPND also belonged to the two Pseudomonadota and Bacteroidota phyla, higher complexity and connectivity of the microbial communities was observed at the genus level compared with samples collected from tanks affected by zoea 2 syndrome. Some species belonging to the Rhodobacteraceae family are potential shrimp probiotics (*Imaizumi et al., 2021*). However, this family has also been reported in diseased cultured shrimp and decreasing in the stomachs of shrimp treated with *Bacillus* probiotic (*Imaizumi et al., 2021*). In the same sense, a higher abundance of the Saprospiraceae family has been reported in the microbiome of shrimp affected by Vibrios (*Zhang et al., 2021*). Though, also members of this family have been described as Vibrio predatory filamentous bacteria (VPFB) that exhibit predation on *V. parahaemolyticus* causing-AHPND, and other *Vibrio* species (*Yeoh et al., 2021*). In our study, two ASVs identified as part of the Saprospiraceae family (including an unidentified genus) were biomarkers significantly more abundant in AHPND-affected larvae.

Vibrios interfere with bacterial networks and change the pattern of bacterial co-occurrence in the stomach of adult *P. vannamei* shrimp. Specifically, complex interactions occur between AHPND and other bacteria, mainly with secondary or opportunist bacteria (*Chen et al., 2017*). For example, AHPND-causing *Vibrio* exhibits positive correlations with commensals *Candidatus Bacilloplasma* and *Cyanobacteria*, indicating a close cooperative interaction with each other at the host microbiome (*Chen et al., 2017*). Consistently, we observed a more complex network of bacteria and a high number of ASVs without taxonomic identification, especially in samples affected by AHPND, indicating a high amount of secondary or opportunist bacteria interacting during the AHPND outbreaks.

Biofilm synthesis is a key factor for the release of PirAB toxins of AHPND-causing bacteria through the gastric sieve into the shrimp hepatopancreas (*Soonthornchai et al., 2015*; *Soowannayan et al., 2019*). This fact increases the probability of pathogens colonization and triggers AHPND (*Tran et al., 2013*; *Dong et al., 2021*). We reported a similar observation in our study, as biofilm formation was a functional biomarker significantly more abundant in AHPND-affected larvae. Also, we found that, valine, leucine, and isoleucine degradation was another functional biomarker significantly more abundant in AHPND-affected larvae. Similarly, the abundance of bacterial genes involved in the degradation of these amino acids complicates the bacterial disease (*Piscirickettsia salmonis*) of *Salmo salar* salmon fish (*Valenzuela-Miranda & Gallardo-Escárate, 2018*). In addition, degradation of these amino acids also occurs in the intestines of adult *P. vannamei* shrimp affected by AHPND (*Cornejo-Granados et al., 2017*). This would likely cause a strong immune response and a change in the gastrointestinal microbiome that ultimately affects the host bacterial communities. It is very plausible that AHPND-causing bacteria were abundant with amino acid degradation and biofilm formation genes that contributed to triggering the disease in the samples from the seven AHPND-positive tanks.

The results generated in this manuscript allow us to understand the microbial profiles of both diseases and are important for the further development of diagnostic tools for zoea 2 syndrome, as the molecular tools for the AHPND diagnostic are well known. In this sense,

parallel to the independent-culture microbiome study described in this manuscript, we collected and preserved samples to study the dependent-culture microbiome. In a further step, we pretend to isolate the pathogens, perform molecular identification, and challenge tests to identify etiological agents causing the zoea 2 syndrome observed in the study. This research was conducted during a critical period when all tanks in the hatchery were affected by *Vibrio* outbreaks, resulting in most tanks having low survival at harvest, and no disease-free tanks were detected. As we did not find tanks with healthy larvae unaffected by AHPND or zoea 2 syndrome it will be necessary to validate the biomarkers found in the study through challenge tests using appropriate experiment controls.

## CONCLUSIONS

We compared the microbiome of *P. vannamei* larvae naturally affected by AHPND and zoea 2 syndrome, and discovered specific microbial signatures that will contribute to the differential diagnostic between both vibriosis conditions. This study reports for the first time the microbiome of larvae affected by zoea 2 syndrome and is a first attempt to compare the microbial profiles of larvae naturally affected by two types of vibriosis. These findings also contribute to the understanding of the microbial ecology mechanisms associated with vibriosis diseases.

### Funding

This study was supported by the Corporación Ecuatoriana para el Desarrollo de la Investigación y la Academia (CEDIA) through its CEPRA program, project CEPRA XV-2021-05 Descubrimiento de Biomarcadores De Potenciales Probióticos Para La Industria Camaronera Ecuatoriana. Guillermo Reyes conducted the research as part of his master's thesis, which was supported by a grant from Secretaría Nacional de Educación Superior, Ciencia, Tecnología e Innovación (SENESCYT). The funders had no role in study design, data collection and analysis, decision to publish, or preparation of the manuscript.

### Grant Disclosures

The following grant information was disclosed by the authors:

Corporación Ecuatoriana para el Desarrollo de la Investigación y la Academia: CEPRA XV-2021-05.

Descubrimiento de Biomarcadores De Potenciales Probióticos Para La Industria Camaronera Ecuatoriana.

Secretaría Nacional de Educación Superior, Ciencia, Tecnología e Innovación (SENESCYT).

### Competing Interests

The authors declare there are no competing interests.

## Author Contributions

- Guillermo Reyes conceived and designed the experiments, performed the experiments, analyzed the data, prepared figures and/or tables, authored or reviewed drafts of the article, and approved the final draft.
- Betsy Andrade performed the experiments, authored or reviewed drafts of the article, and approved the final draft.
- Irma Betancourt performed the experiments, authored or reviewed drafts of the article, and approved the final draft.
- Fanny Panchana performed the experiments, authored or reviewed drafts of the article, and approved the final draft.
- Ramiro Solórzano performed the experiments, authored or reviewed drafts of the article, and approved the final draft.
- Cristhian Preciado performed the experiments, authored or reviewed drafts of the article, and approved the final draft.
- Lita Sorroza performed the experiments, authored or reviewed drafts of the article, and approved the final draft.
- Luis E. Trujillo performed the experiments, authored or reviewed drafts of the article, and approved the final draft.
- Bonny Bayot conceived and designed the experiments, performed the experiments, analyzed the data, authored or reviewed drafts of the article, and approved the final draft.

## Data Availability

The sequences (derived from NovaSeq P250 sequencing) are available at GenBank: PRJNA870476.

## Supplemental Information

Supplemental information for this article can be found online at http://dx.doi.org/10.7717/peerj.15795#supplemental-information.

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
