# Peer review of "Microbial signature profiles of Penaeus vannamei larvae in low-survival hatchery tanks affected by vibriosis"

_PeerJ, doi:10.7717/peerj.15795_

## Round 0.1 · original submission · Major Revisions

Your manuscript has to be deeply improved. It is very important to add some information regarding the use of control samples in the experiments. Furthermore, the criteria for determining other vibriosis are relevant and should be also included.

Reviewer 1 ·

Basic reporting

This is an interesting manuscript on vibriosis in shrimp. The manuscript text is generally clear but there are several grammar errors present in the work, such as incorrect use of the term 'while' and some confusion regarding some sentences. While minor, a full proofread would help to reduce these.
There is an excellent base of references provided and they are well used in the work. However, it should be noted that these have not been formatted correctly to the PeerJ style and so full revisions are required. The context is provided, but the message in the introduction is confused. In the discussion there is reference to another published paper by the authors in the same study field and at current it is unclear how the two studies are interlinked. This needs to be signposted more clearly.
The raw data seem to be available in the form of supplementary materials, though this should be made clearer with a data availability statement.

Experimental design

The experimental design could be explained more clearly. There is mention of 48 samples from 12 tanks, but the sample selection and larval stages could be elucidated more clearly. My concern is the lack of control to the study - as there is no control (vibrio-free shrimp) it is unclear what the shrimp microbiome should be like. There is also no ethical statement and while invertebrates are not typically protected, I would have liked to see some consideration of sample size and euthanasia.
This said, the sampling methods provided are clearly explained.

Validity of the findings

Overall, this is a meaningful study and there could be some valid and useful findings. At current, the wider impact of the study is confused as a result of the lack of control and the questions surrounding where this paper fits alongside previously published research. If the manuscript is to proceed, both these points need to be addressed clearly. Without these points, it is difficult for the authors to draw some of the conclusions that are raised in the discussion and conclusion.

Annotated reviews are not available for download in order to protect the identity of reviewers who chose to remain anonymous.

Reviewer 2 ·

Basic reporting

“Diseases” was clearly identified by the farmer as the top challenge faced in shrimp farming. Vibriosis is one of the most common diseases, which caused by some pathogenic Vibrio, such as Vibrio alginolyticus, Vibrio parahaemolyticus and Vibrio harveyi. In this study, the authors compared the microbiome of Penaeus vannamei larvae affected by AHPND and other vibriosis. The results of this manuscript will help us to well know the microbial composition of Penaeus vannamei larvae after affected by different vibrio species. Several questions should be clarified before acceptance.

1. In this manuscript, two disease conditions have been compared, it can be found the differences in the microbiome of Penaeus vannamei larvae between two groups. Why did not the author add the normal group (healthy larvae) in this manuscript? That will be more significant if the authors add the normal group.
2. Are Penaeus vannamei larvae from the same pair of parents?
3. “other vibriosis”? What is the criterion of the other vibriosis? I suggest the authors add the relevant information in the manuscript.
4. The authors should add the information of the other vibriosis, such as the potential symptoms. In addition, the authors should explain which vibriosis species cause the “other vibriosis”.
5. The authors should supplement feed information and rearing water conditions at the part of Materials and Methods.
6. Line 228, 233, 235, and so on. “three-fold”, “two-fold”, “seven-fold”? Is there significant difference between AHPND group and the other vibriosis group? The authors should further spell out in the manuscript.
7. Line 360-363: The co-occurrence network has been analyzed at the phylum and genus levels. There is no relevant information on family level. So, how did the authors get the conclusion that higher complexity and connectivity of the microbial communities was observed at the family level compared with samples collected from tanks affected by other vibriosis.

Experimental design

Generally, research question is well defined in this manuscript. The experimental methods are acceptable.

Validity of the findings

no comment

Reviewer 3 ·

Basic reporting

No comments

Experimental design

2.1. The authors distinguish between AHPND and other vibriosis, but is there a justification for lumping all other vibriosis types as a single category? Could this grouping be a contributing factor to the lack of significant difference between microbiome compositions by disease state?

2.2. If the pVA1 plasmid encoding the disease-causing toxin can be readily transferred between Vibrio spp., does it make sense to characterize microbiome signatures at a deeper taxonomic level?

2.3. Samples infected with viral shrimp pathogens were discarded and excluded from analysis. While this may go beyond the scope of this manuscript, it may be beneficial to construct microbiome profiles for these virally-infected populations as well.

Validity of the findings

No comments

Additional comments

While a few minor point may deserve further clarification and discussion, the manuscript is sound overall, and contributes to the study of shrimp microbiomes.

---

## Round 0.2 · accepted · Accept

Thank you for submitting your work to this journal.

With kind regards,

Reviewer 1 ·

Basic reporting

The revised manuscript provides a much clearer explanation of the background of the study, and the wider implications and rationale to the research The manuscript text is generally clear and original grammatical errors have been addressed.
There is an excellent base of references provided and they are well used in the work. However, it should be noted that these have not been formatted correctly to the PeerJ style and this needs adjusting still. It should be noted this was commented on during the initial review..
The raw data points from the initial review have been well addressed.

Experimental design

The lack of control concern is still valid as a result of methodological design. This has now been acknowledged as a limitation in the study's discussion and highlighted as a future area for development. The study is now better defined and the methods are clearer as a result of the revisions. Points pertaining to number of samples are now more clearly explained.

Validity of the findings

Initial concerns relating to reference to previous studies have been well addressed during the revision process. There is now reference to the lack of control briefly before the conclusion, though this could be elaborated on a little further to provide context. This said, the conclusions are well in line with the findings highlighted in this study.

Additional comments

Dear Authors,
Thank you for providing a revised version of your manuscript, alongside a rebuttal letter. Your revisions were clearly to follow and address the majority of the concerns raised from the initial review of the manuscript.

Reviewer 3 ·

Basic reporting

The authors addressed all previous comments, and have improved the structure and text of the manuscript.

Experimental design

The authors have addressed all previous comments.

Validity of the findings

The author responses and added information clarified all previous comments regarding these aspects of the work.